# DetNAS: Backbone Search for Object Detection

**Yukang Chen**[1][†][*], **Tong Yang**[2][†], **Xiangyu Zhang**[2][‡], **Gaofeng Meng**[1], **Xinyu Xiao**[1], **Jian Sun**[2]
[1]*National Laboratory of Pattern Recognition, Institute of Automation, Chinese Academy of Sciences*
[2]*Megvii Technology*
{yukang.chen, gfmeng, xinyu.xiao}@nlpr.ia.ac.cn  {yangtong, zhangxiangyu, sunjian}@megvii.com

## Abstract

Object detectors are usually equipped with backbone networks designed for image classification. It might be sub-optimal because of the gap between the tasks of image classification and object detection. In this work, we present DetNAS to use Neural Architecture Search (NAS) for the design of better backbones for object detection. It is non-trivial because detection training typically needs ImageNet pre-training while NAS systems require accuracies on the target detection task as supervisory signals. Based on the technique of one-shot supernet, which contains all possible networks in the search space, we propose a framework for backbone search on object detection. We train the supernet under the typical detector training schedule: ImageNet pre-training and detection fine-tuning. Then, the architecture search is performed on the trained supernet, using the detection task as the guidance. This framework makes NAS on backbones very efficient. In experiments, we show the effectiveness of DetNAS on various detectors, for instance, one-stage RetinaNet and the two-stage FPN. We empirically find that networks searched on object detection shows consistent superiority compared to those searched on ImageNet classification. The resulting architecture achieves superior performance than hand-crafted networks on COCO with much less FLOPs complexity. Code and models have been made available at: https://github.com/megvii-model/DetNAS.

## 1 Introduction

Backbones play an important role in object detectors. The performance of object detectors highly relies on features extracted by backbones. For example, a large accuracy increase could be obtained by simply replacing a ResNet-50 [8] backbone with stronger networks, e.g., ResNet-101 or ResNet-152. The importance of backbones is also demonstrated in DetNet [12], Deformable ConvNets v2 [30], ThunderNet [22] in real-time object detection, and HRNet [25] in keypoint detection.

However, many object detectors directly use networks designed for Image classification as backbones. It might be sub-optimal because image classification focus on *what* the main object of an image is, while object detection aims at finding *where and what* each object instance. For instance, the recent hand-crafted network, DetNet [12], has demonstrated this point. ResNet-101 performs better than DetNet-59 [12] on ImageNet classification, but is inferior to DetNet-59 [12] on object detection. However, the handcrafting process heavily relies on expert knowledge and tedious trials.

NAS has achieved great progress in recent years. On image classification [31, 32, 23], searched networks reach or even surpass the performance of the hand-crafted networks. However, NAS for backbones in object detectors is still challenging. It is infeasible to simply employ previous NAS methods for backbone search in object detectors. The typical detector training schedule requires backbone networks to be pre-trained on ImageNet. This results in two difficulties for searching

---

[†]Equal contribution.  [*]Work done during an internship at Megvii Technology. [‡] Corresponding author.

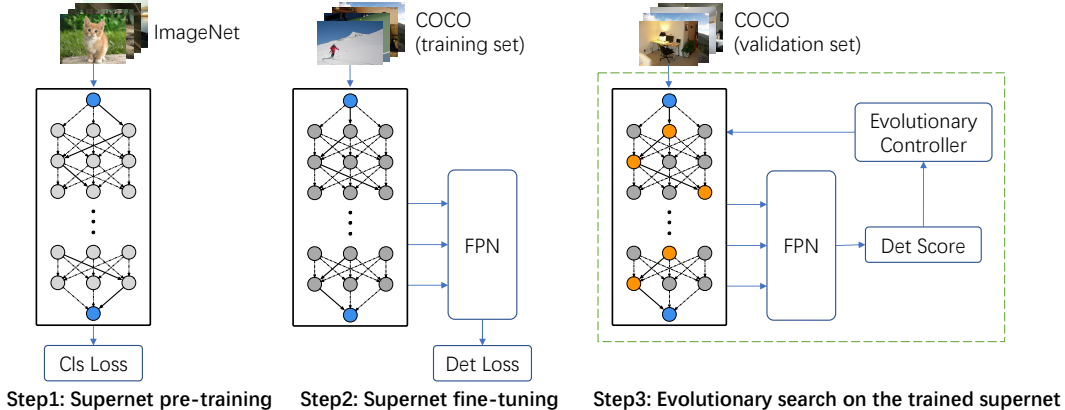

Step1: Supernet pre-training    Step2: Supernet fine-tuning    Step3: Evolutionary search on the trained supernet

Figure 1: The pipeline of DetNAS that searches for backbones in object detectors. There are three steps: supernet pre-training on ImageNet, supernet fine-tuning on the detection training set, e.g., COCO, and architecture search on the trained supernet with the evolution algorithm. The validation set is actually split from COCO `trainval35k` and consists of 5k images.

backbones in object detectors: 1) *hard to optimize*: NAS systems require accuracies on target tasks as reward signals, pre-training accuracy is unqualified for this requirement; 2) *inefficiency*: In order to obtain the precious performance, each candidate architecture during search has to be first pretrained (e.g. on ImageNet) then finetuned on the detection dataset, which is very costly. Even though training from scratch is an alternative [10], it requires more training iterations to compensate for the lack of pre-training. Moreover, training from scratch breaks down in small datasets, e.g. PASCAL VOC.

In this work, we present the first effort on searching for *backbones* in object detectors. Recently, a NAS work on object detection, NAS-FPN [5], is proposed. It searches for feature pyramid networks (FPN) [14] rather than backbones. It can perform with a pre-trained backbone network and search with the previous NAS algorithm [31]. Thus, the difficulty of backbone search is still unsolved. Inspired by one-shot NAS methods [7, 1, 2], we solve this issue by decoupling the weight training and architecture search. Most previous NAS methods optimize weights and architectures in a nested manner. Only if we decouple them into two stages, the pre-training step can be incorporated economically. This framework avoids the inefficiency issue caused by the pre-training and makes backbone search feasible.

The framework of DetNAS consists of three steps: (1) pre-training the one-shot supernet on ImageNet, (2) fine-tuning the one-shot supernet on detection datasets, (3) architecture search on the trained supernet with an evolutionary algorithm (EA). In experiments, the main result backbone network, DetNASNet, with much fewer FLOPs, achieves 2.9% better mmAP than ResNet-50 on COCO with the FPN detector. Its enlarged version, DetNASNet (3.8), is superior to ResNet-101 by 2.0% on COCO with the FPN detector. In addition, we validate the effectiveness of DetNAS in different detectors (the two-stage FPN [14] and the one-stage RetinaNet [15]) and various datasets (COCO and VOC). DetNAS are consistently better than the network searched on ImageNet classification by more than 3% on VOC and 1% on COCO, no matter on FPN or RetinaNet.

Our main contributions are summarized as below:

- We present DetNAS, a framework that enables backbone search for object detection. To our best knowledge, this is the first work on this challenging task.

- We introduce a powerful search space. It helps the searched networks obtain inspiring accuracies with limited FLOPs complexity.

- Our result networks, DetNASNet and DetNASNet (3.8), outperforms the hand-crafted networks by a large margin. Without the effect of search space, we show the effectiveness of DetNAS in different detectors (two-stage FPN and one-stage RetinaNet), various datasets (COCO and VOC). The searched networks have consistently better performance and meaningful structural patterns.

## 2 Related Work

### 2.1 Object Detection

Object detection aims to locate each object instance and assign a class to it in an image. With the rapid progress of deep convolutional networks, object detectors, such as FPN [14] and RetinaNet [15], have achieved great improvements in accuracy. In general, an object detector can be divided into two parts, a backbone network, and a "head". In the past few years, many advances in object detection come from the study of "head", such as architecture [14], loss [15, 24], and anchor [29, 26]. FPN [14] develops a top-down architecture with lateral connections to integrate features at all scales as an effective feature extractor. The focal loss [15] is proposed in RetinaNet to solve the problem of class imbalance, which leads to the instability in early training. MetaAnchor [29] proposes a dynamic anchor mechanism to boost the performance for anchor-based object detectors. However, for the backbone network, almost all object detectors adopt networks for image classification, which might be sub-optimal. Because object detection cares about not only "what" object is, which image classification only focuses, but also "where" it is. Similar to our work, DetNet [12] also exploits the architecture of the backbone that specially designed for object detection manually. Inspired by NAS, we present DetNAS to find the optimal backbone automatically for object detection in this work.

### 2.2 Neural Architecture Search

**NAS on image classification** Techniques to design networks automatically have attracted increasing research interests. NAS [31] and NASNet [32] use reinforcement learning (RL) to determine neural architectures sequentially. In addition to these RL-based methods, the evolution algorithm (EA) also shows its potential. AmeobaNet [23] proves that the basic evolutionary algorithm without any controller can also achieve comparable results and even surpass RL-base methods. To save computational resources, some works propose to use weight sharing or one-shot methods, e.g., ENAS [21] and DARTS [17]. Many following works, including SNAS [28], Proxyless [3] and FBNet [27] and others [4], also belong to one-shot NAS to some extent.

**NAS on other tasks** In addition to NAS works on image classification, some recent works attempt to develop NAS to other tasks, especially semantic segmentation. [19] proposes to search auxiliary cells as the segmentation decoders. Auto-DeepLab [16] applies the gradient-based method to search backbones for segmentation models. To our best knowledge, no works have attempted to search neural architectures for backbones in object detectors. One main reason might come from the costly ImageNet pre-training for object detectors. Training from scratch scheme [10], as a substitute, proves to bring no computational savings and tends to break down in small datasets. In this work, we overcome this obstacle with a one-shot supernet and the evolutionary search algorithm.

## 3 Detection Backbone Search

Our goal is to extend NAS to search for backbones in object detectors. In general, object detector training typically requires ImageNet pre-training. Meanwhile, NAS systems require supervisory signals from target tasks. For each network candidate, it needs ImageNet pre-training, which is computationally expensive. Additionally, training from scratch is an alternative method while it requires more iterations to optimize and breaks down in small datasets. Inspired by the one-shot NAS [7, 2, 1], we decouple the one-shot supernet training and architecture optimization to overcome this obstacle. In this section, we first clarify the motivation of our methodology.

### 3.1 Motivation

Without loss of generality, the architecture search space $\mathcal{A}$ can be denoted by a directed acyclic graph (DAG). Any path in the graph corresponds to a specific architecture, $a \in \mathcal{A}$. For the specific architecture, its corresponding network can be represented as $\mathcal{N}(a, w)$ with the network weights $w$. NAS aims to find the optimal architecture $a^* \in \mathcal{A}$ that minimizes the validation loss $\mathcal{L}_{val}(\mathcal{N}(a^*, w^*))$. $w^*$ denotes the optimal network weights of the architecture $a^*$. It is obtained by minimizing the

training loss. We can formulate NAS process as a nested optimization problem:

$$\min_{a \in \mathcal{A}} \mathcal{L}_{val}(\mathcal{N}(a, w^*(a))) \tag{1}$$

$$\text{s.t. } w^*(a) = \arg\min_{w} \mathcal{L}_{train}(w, a) \tag{2}$$

The above formulation can represent NAS on tasks that work without pre-training, e.g., image classification. But for object detection, which needs pre-training and fine-tuning schedule, Eq. (2) needs to be reformulated as follow:

$$w^*(a) = \arg\min_{w \leftarrow w_p(a)^*} \mathcal{L}_{train}^{det}(w, a) \quad \text{s.t. } w_p(a)^* = \arg\min_{w_p} \mathcal{L}_{train}^{cls}(w_p, a) \tag{3}$$

where $w \leftarrow w_p(a)^*$ is to optimize $w$ with $w_p(a)^*$ as initialization. The pre-trained weights $w_p(a)^*$ can not directly serve for the Eq. (1), but it is necessary for $w(a)^*$. Thus, we can not skip the ImageNet pre-training in DetNAS. However, ImageNet pre-training usually costs several GPU days just for a single network. It is unaffordable to train all candidate networks individually. In one-shot NAS methods [7, 2, 1], the search space is encoded in a supernet which consists of all candidate architectures. They share the weights in their common nodes. In this way, Eq. (1) and Eq. (2) become:

$$\min_{a \in \mathcal{A}} \mathcal{L}_{val}(\mathcal{N}(a, W_{\mathcal{A}}^*(a))) \quad \text{s.t. } W_{\mathcal{A}}^* = \arg\min_{W} \mathcal{L}_{train}(\mathcal{N}(\mathcal{A}, W)) \tag{4}$$

where all individual network weights $w(a)$ are inherited from the one-shot supernet $W_{\mathcal{A}}$. The supernet training, $W_{\mathcal{A}}$ optimization, is decoupled from the architecture $a$ optimization. Based on this point, we go further step to incorporate pre-training step. This enables NAS on the more complicated task, backbone search in object detecion:

$$\min_{a \in \mathcal{A}} \mathcal{L}_{val}^{det}(\mathcal{N}(a, W_{\mathcal{A}}^*(a))) \tag{5}$$

$$\text{s.t. } W_{\mathcal{A}}^* = \arg\min_{W \leftarrow W_{p\mathcal{A}}^*} \mathcal{L}_{train}^{det}(\mathcal{N}(\mathcal{A}, W)) \tag{6}$$

$$W_{p\mathcal{A}}^* = \arg\min_{W_p} \mathcal{L}_{train}^{cls}(\mathcal{N}(\mathcal{A}, W_p)) \tag{7}$$

## 3.2 Our NAS Pipeline

As in Fig. 1, DetNAS consists of 3 steps: supernet pre-training on ImageNet, supernet fine-tuning on detection datasets and architecture search on the trained supernet.

**Step 1: Supernet pre-training.** ImageNet pre-training is the fundamental step of the fine-tuning schedule. For some one-shot methods [17, 27], they relax the actually discrete search space into a continuous one, which makes the weights of individual networks deeply coupled. However, in supernet pre-training, we adopt a path-wise [7] manner to ensure the trained supernet can reflect the relative performance of candidate networks. Specifically, in each iteration, only one single path is sampled for feedforward and backward propagation. No gradient or weight update acts on other paths or nodes in the supernet graph.

**Step 2: Supernet fine-tuning.** The supernet fine-tuning is also path-wise but equipped with detection head, metrics and datasets. The other necessary detail to mention is about batch normalization (BN). BN is a popular normalization method to help optimization. Typically, the parameters of BN, during fine-tuning, are fixed as the pre-training batch statistics. However, the freezing BN is infeasible in DetNAS, because the features to normalize are not equal on different paths. On the other hand, object detectors are trained with high-resolution images, unlike image classification. This results in small batch sizes as constrained by memory and severely degrades the accuracy of BN. To this end, we replace the conventional BN with Synchronized Batch Normalization (SyncBN) [20] during supernet training. It computes batch statistics across multiple GPUs and increases the effective batch size. We formulate the supernet training process in Algorithm 1 in the supplementary material.

**Step 3: Search on supernet with EA.** The third step is to conduct the architecture search on the trained supernet. Paths in the supernet are picked and evaluated under the direction of the evolutionary controller. For the evolutionary search, please refer to Section 3.4 for details. The necessary detail in this step is also about BN. During search, different child networks are sampled path-wise in the

Table 1: Search space of DetNAS.

| Stage | Block | Large (40 blocks) | | Small (20 blocks) | |
|---|---|---|---|---|---|
| | | $c_1$ | $n_1$ | $c_2$ | $n_2$ |
| 0 | Conv3×3-BN-ReLU | 48 | 1 | 16 | 1 |
| 1 | ShuffleNetv2 block (search) | 96 | 8 | 64 | 4 |
| 2 | ShuffleNetv2 block (search) | 240 | 8 | 160 | 4 |
| 3 | ShuffleNetv2 block (search) | 480 | 16 | 320 | 8 |
| 4 | ShuffleNetv2 block (search) | 960 | 8 | 640 | 4 |

* ShuffleNetv2 block has 4 choices for search: 3x3, 5x5, 7x7 and Xception 3x3.

supernet. The issue is that the batch statistics on one path should be independent of others. Therefore, we need to *recompute batch statistics* for each single path (child networks) before each evaluation. This detail is indispensable in DetNAS. We extract a small subset of the training set (500 images) to recompute the batch statistics for the single path to be evaluated. This step is to accumulate reasonable running mean and running variance values for BN. It involves no gradient backpropagation.

### 3.3 Search Space Design

The details about the search space are described in the Table 1. Our search space is based on the ShuffleNetv2 block. It is a kind of efficient and lightweight convolution architectures and involves channel split and shuffle operation [18]. We design two search spaces with different sizes, the large one for the main result and the small one for ablation studies.

**Large (40 blocks)** This search space is a large one and designed for the main results to compare with hand-crafted backbone networks. The channels and blocks in each stage are specified by $c_1$ and $n_1$. In each stage, the first block has stride 2 for downsampling. Except for the first stage, there are 4 stages that contain $8 + 8 + 16 + 8 = 40$ blocks for search. For each block to search, there are 4 choices developed from the original ShuffleNetv2 block: changing the kernel size with {3×3, 5×5, 7×7} or replacing the right branch with an Xception block (three repeated separable depthwise 3×3 convolutions). It is easy to count that this search space includes $4^{40} \approx 1.2 \times 10^{24}$ candidate architectures. Most networks in this search space have more than 1G FLOPs. We construct this large search space is for the comparisons with hand-crafted large networks. For example, ResNet-50 and ResNet-101 have 3.8G and 7.6G FLOPs respectively.

**Small (20 blocks)** This search space is smaller and designed for ablation studies. The channels and blocks in each stage are specified by $c_2$ and $n_2$. The block numbers $n_1$ are twice as $n_2$. The channel numbers $c_1$ are 1.5 times as $c_2$ in all searched stages. This search space includes $4^{20} \approx 1.0 \times 10^{12}$ possible architectures. It is still a large number and enough for ablation studies. Most networks in this search space have around 300M FLOPs. We conduct all situation comparisons in this search space, including various object detectors (FPN or RetinaNet), different datasets (COCO or VOC), and different schemes (training from scratch or with pre-training).

### 3.4 Search Algorithm

The architecture search step is based on the evolution algorithm. At first, a population of networks **P** is initialized randomly. Each individual $P$ consists of its architecture $P.\theta$ and its fitness $P.f$. Any architecture against the constraint $\eta$ would be removed and a substitute would be picked. After initialization, we start to evaluate the individual architecture $P.\theta$ to obtain its fitness $P.f$ on the validation set $\mathbb{V}_{Det}$. Among these evaluated networks, we select the top $|\mathcal{P}|$ as $parents$ to generate $child$ networks. The next generation networks are generated by $mutation$ and $crossover$ half by half under the constraint $\eta$. By repeating this process in iterations, we can find a single path $\theta_{best}$ with the best validation accuracy or fitness, $f_{best}$. We formulate this process as Algorithm 2 in the supplementary material. The hyper-parameters in this step are introduced in Section 4.

Compared to RL-based [32, 31, 21] and gradient-based NAS methods [17, 27, 3], the evolutionary search can stably meet hard constraints, e.g., FLOPs or inference speed. To optimize FLOPs or inference speed, RL-based methods need a carefully tuned reward function while gradient-based methods require a wise designed loss function. But their outputs are still hard to totally meet the required constraints. To this end, DetNAS chooses the evolutionary search algorithm.

Table 2: Main result comparisons.

| Backbone | ImageNet Classification | | Object Detection with FPN on COCO | | | | | |
|---|---|---|---|---|---|---|---|---|
| | FLOPs | Accuracy | mAP | $AP_{50}$ | $AP_{75}$ | $AP_s$ | $AP_m$ | $AP_l$ |
| ResNet-50 | 3.8G | 76.15 | 37.3 | 58.2 | 40.8 | 21.0 | 40.2 | 49.4 |
| ResNet-101 | 7.6G | 77.37 | 40.0 | 61.4 | 43.7 | 23.8 | 43.1 | 52.2 |
| ShuffleNetv2-40 | 1.3G | 77.18 | 39.2 | 60.8 | 42.4 | 23.6 | 42.3 | 52.2 |
| ShuffleNetv2-40 (3.8) | 3.8G | 78.47 | 40.8 | 62.1 | 44.8 | 23.4 | 44.2 | 54.2 |
| DetNASNet | 1.3G | 77.20 | 40.2 | 61.5 | 43.6 | 23.3 | 42.5 | 53.8 |
| DetNASNet (3.8) | 3.8G | 78.44 | **42.0** | 63.9 | 45.8 | 24.9 | 45.1 | 56.8 |

\* These are trained with the same "1x" settings in Section 4. The "2x" results are in the supplementary material.

# 4   Experimental Settings

**Supernet pre-training.**   For ImageNet classification dataset, we use the commonly used 1.28M training images for supernet pre-training. To train the one-shot supernet backbone on ImageNet, we use a batch size of 1024 on 8 GPUs for 300k iterations. We set the initial learning rate to be 0.5 and decrease it linearly to 0. The momentum is 0.9 and weight decay is $4 \times 10^{-5}$.

**Supernet fine-tuning.**   We validate our method with two different detectors. The training images are resized such that the shorter size is 800 pixels. We train on 8 GPUs with a total of 16 images per minibatch for 90k iterations on COCO and 22.5k iterations on VOC. The initial learning rate is 0.02 which is divided by 10 at {60k, 80k} iterations on COCO and {15k, 20k} iterations on VOC. We use weight decay of $1 \times 10^{-4}$ and momentum of 0.9. For the head of FPN, we replace two fully-connected layers (2fc) with 4 convolutions and 1 fully-connected layer (4conv1fc), which is also used in all baselines in this work, e.g., ResNet-50, ResNet-101, ShuffleNetv2-20 and ShuffleNetv2-40. For RetinaNet, the training settings is similar to FPN, except that the initial learning rate is 0.01.

**Search on the trained supernet.**   We split the detection datasets into a training set for supernet fine-tuning, a validation set for architecture search, and a test set for final evaluation. For VOC, the validation set contains 5k images randomly selected from trainval2007 + trainval2012 and the remains for supernet fine-tuning. For COCO, the validation set contains 5k images randomly selected from trainval35k [13] and the remains for supernet fine-tuning. For evolutionary search, the evolution process is repeated for 20 iterations. The population size $|\mathbf{P}|$ is 50 and the parents size $|\mathcal{P}|$ is 10. Thus, there are 1000 networks evaluated in one search.

**Final architecture evaluation.**   The selected architectures are also retrained in the pre-training and fine-tuning schedule. The training configuration is the same as that of the supernet. For COCO, the test set is minival. For VOC, the test set is test2007. Results are mainly evaluated with COCO standard metrics (i.e. mmAP) and VOC metric (IOU=.5). All networks listed in the paper are trained with the "1x" training setting used in Detectron [6] to keep consistency with the supernet fine-tuning.

# 5   Experimental Results

## 5.1   Main Results

Our main result architecture, DetNASNet, is searched on FPN in the large search space. The architecture of DetNASNet is depicted in the supplementary material. We search on FPN because it is a mainstream two-stage detector that has been used in other vision tasks, e.g., instance segmentation and skeleton detection [9]. Table 2 shows the main results. We list three hand-crafted networks for comparisons, including ResNet-50, ResNet-101 and ShuffleNetv2-40. DetNASNet achieves 40.2% mmAP with only 1.3G FLOPs. It is superior to ResNet-50 and equal to ResNet-101.

To eliminate the effect of search space, we compare with the hand-crafted ShuffleNetv2-40, a baseline in this search space. It has 40 blocks and is scaled to 1.3G FLOPs, which are identical to DetNASNet. ShuffleNetv2-40 is inferior to ResNet-101 and DetNAS by 0.8% mmAP on COCO. This shows the effectiveness of DetNAS without the effect of search space.

Table 3: Ablation studies.

| | ImageNet (Top1 Acc, %) | COCO (mmAP, %) | | VOC (mAP, %) | |
|---|---|---|---|---|---|
| | | FPN | RetinaNet | FPN | RetinaNet |
| ShuffleNetv2-20 | 73.1 | 34.8 | 32.1 | 80.6 | 79.4 |
| ClsNASNet | **74.3** | 35.1 | 31.2 | 78.5 | 76.5 |
| DetNAS-scratch | 73.8 - 74.3 | 35.9 | 32.8 | 81.1 | 79.9 |
| DetNAS | 73.9 - 74.1 | **36.6** | **33.3** | **81.5** | **80.1** |

\* DetNAS and DetNAS-scratch both have 4 specific networks in each case (COCO/VOC, FPN/RetinaNet). The ImageNet classification accuracies of DetNAS and DetNAS-scratch are the minimal to the maximal.

Table 4: Computation cost for each step on COCO.

| | Supernet pre-training | Supernet fine-tuning | Search on the supernet |
|---|---|---|---|
| DetNAS | $3 \times 10^5$ iterations | $9 \times 10^4$ iterations | $20 \times 50$ models |
| | 8 GPUs on 1.5 days | 8 GPUs on 1.5 days | 20 GPUs on 1 day |

\* For the small search space, GPUs are GTX 1080Ti . For the large search space, GPUs are Tesla V100.

After that, we include the effect of search space for consideration and increase the channels of DetNASNet by 1.8 times to 3.8G FLOPs, DetNASNet (3.8). Its FLOPs are identical to that of ResNet-50. It achieves 42.0% mmAP which surpasses ResNet-50 by 4.7 % and ResNet-101 by 2.0%.

## 5.2 Ablation Studies

The ablation studies are conducted in the small search space introduced in Table 1. This search space is much smaller than the large one, but it is efficient and enough for making ablation studies. As in Table 3, we validate the effectiveness of DetNAS with various detectors (FPN and RetinaNet) and datasets (COCO and VOC). All models in Table 3 and Table 5 are trained with the same settings described in Section 4. Their FLOPs are all similar and under 300M.

**Comparisons to the hand-crafted network.**
ShuffleNetv2-20 is a baseline network constructed with 20 blocks and scaled to 300M FLOPs. It has the same number of blocks to architectures searched in the search space. As in Table 3, DetNAS shows a consistent superiority to the hand-crafted ShuffleNetv2-20. DetNAS outperforms ShuffleNetv2-20 by more than 1% mmAP in COCO on both FPN and RetinaNet detectors. This shows that NAS in object detection can also achieve a better performance than the hand-crafted network.

**Comparisons to the network for ImageNet classification.**
Some NAS works tend to search on small proxy tasks and then transfer to other tasks or datasets. For example, NASNet is searched from CIFAR-10 image classification and directly applied to object detection [32]. We empirically show this manner sub-optimal. ClsNASNet is the best architecture searched on ImageNet classification. The search method and search space follow DetNAS. We use it as the backbone of object detectors. ClsNASNet is the best on ImageNet classification in Table 3 while its performance on object detection is disappointing. It is the worst in all cases, except a slightly better than ShuffleNetv2-20 on COCO-FPN. This shows that NAS on target tasks can perform better than NAS on proxy tasks.

**Comparisons to the from-scratch counterpart.**
DetNAS-scratch is a from-scratch baseline to ablate the effect of pre-training. In this case, the supernet is trained from scratch on detection datasets without being pre-trained on ImageNet. To compensate for the lack of pretraining, its training iterations on detection datasets are twice as those of DetNAS, that is, 180k on COCO and 45k on VOC. In this way, the computation cost of DetNAS and DetNAS-scratch are similar. All other settings are the same as DetNAS. Both DetNAS and DetNAS-scratch have consistent improvements on ClsNASNet and ShuffleNetv2-20. This shows that searching directly on object detection is a better choice, no matter from scratch or with ImageNet pre-training. In addition, DetNAS performs also better than DetNAS-scratch in all cases, which reflects the importance of pre-training.

Table 5: Comparisons to the random baseline.

| | ImageNet (Top1 Acc, %) | COCO (mmAP, %) | | VOC (mAP, %) | |
|---|---|---|---|---|---|
| | | FPN | RetinaNet | FPN | RetinaNet |
| Random | $73.9 \pm 0.2$ | $35.6 \pm 0.6$ | $32.5 \pm 0.4$ | $80.9 \pm 0.2$ | $79.0 \pm 0.7$ |
| DetNAS | 73.9 - 74.1 | 36.6 | 33.3 | 81.5 | 80.1 |

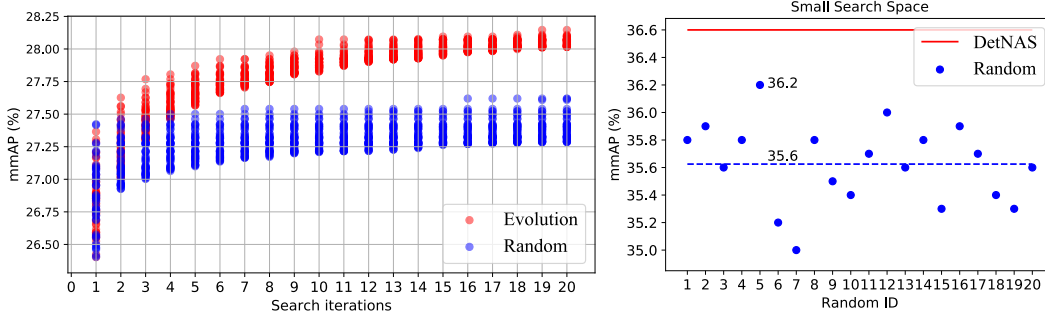

Figure 2: Curve of EA and Random during search.  Figure 3: Random models on COCO-FPN.

**Comparisons to the random baseline.**
As stated in many NAS works [11, 17], the random baseline is also competit ive. In this work, we also include the random baseline for comparisons as in Table 5. In Figure 2, the mmAP curve on the supernet search are depicted to compare EA with Random. For each iteration, top 50 models until the current iteration are depicted at each iteration. EA demonstrates a clearly better sampling efficiency than Random. In addition, we randomly pick 20 networks in the search space and train them with the same settings to other result models. On ImageNet classification, the random baseline is comparable to DetNAS. But on the object detection tasks, DetNAS performs better than Random. In Figure 3, we depicted the scatter and the average line of random models and the line of DetNAS. DetNAS in the small search space is 36.6% while the Random is 35.6±0.6%. From these points of view, DetNAS performs better than Random not only during search but also in the output models.

## 5.3 DetNAS Architecture and Discussions

Our architectures searched for object detectors show meaningful patterns that are distinct from architectures searched for image classification. Figure 4 illustrates three neural architectures searched in the 20 blocks search space. The architecture in the top of Figure 4 is ClsNASNet. The other two are searched with FPN and RetinaNet detectors respectively. These architectures are depicted block-wise. The yellow and orange blocks are 5×5 and 7×7 ShuffleNetv2 blocks. The blue blocks have kernel sizes as 3. The larger blue blocks are Xception ShuffleNetv2 blocks which are deeper than the small 3×3 ShuffleNetv2 blocks. Figure 5 illustrates the architecture of DetNASNet. It has 40 blocks in total and {8, 8, 16, 8} blocks each stage.

In contrast to ClsNASNet, architectures of DetNAS have large-kernel blocks in low-level layers and deep blocks in high-level layers. In DetNAS, blocks of large kernels (5×5, 7×7) almost gather in low-level layers, Stage 1 and Stage 2. In contrast, ClsNASNet has all 7×7 blocks in Stage 3 and Stage 4. This pattern also conforms to the architectures of ProxylessNAS [3] that is also searched on image classification and ImageNet dataset. On the other hand, blocks with blue color have 3×3 kernel size. As in the middle of Figure 4, blue blocks are almost grouped in Stage 3 and Stage 4. Among these 8 blue blocks, 6 blocks are Xception ShufflNetv2 blocks that are deeper than common 3×3 ShufflNetv2 blocks. In ClsNASNet, only one Xception ShuffleNetv2 block exists in high-level layers. In addition, DetNASNet also shows the meaningful pattern that most high-levels blocks have 3×3 kernel size. Based on these observations, we find that the networks suitable for object detection are visually different from the networks for classification. Therefore, these distinctions further confirm the necessity of directly searching on the target tasks, instead of proxy tasks.

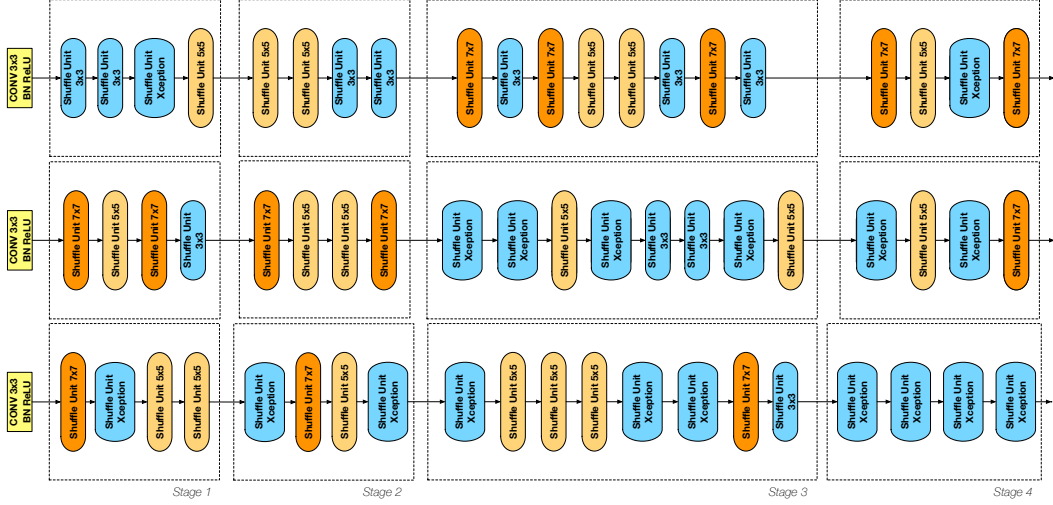

Figure 4: The searched architecture pattern comparison in the small (20 blocks) search space. From top to bottom, they are ClsNASNet, DetNAS (COCO-FPN) and DetNAS (COCO-RetinaNet).

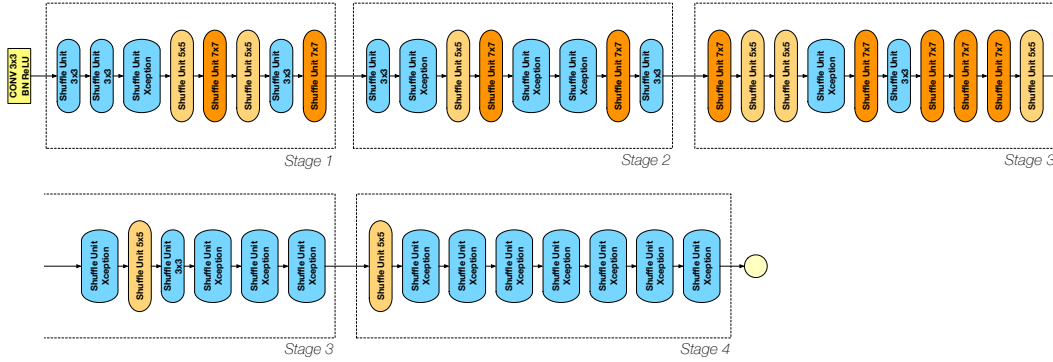

Figure 5: DetNASNet architecture

# 6 Conclusion

We present DetNAS, the first attempt to search backbones in object detectors without any proxy. Our method consists of three steps: supernet pre-training on ImageNet, supernet fine-tuning on detection datasets and searching on the trained supernet with EA. Table 4 shows the computation cost for each steps. The computation cost of DetNAS, 44 GPU days on COCO, is just two times as training a common object detector. In experiments, the main result of DetNAS achieves superior performance than ResNet-101 on COCO and the FPN detector with much fewer FLOPs complexity. We also make comprehensive comparisons to show the effectiveness of DetNAS. We test DetNAS on various object detectors (FPN and RetinaNet) and different datasets (COCO and VOC). For further discussions, we spotlight the architecture-level gap between image classification and object detection. ClsNASNet and DetNAS have different and meaningful architecture-level patterns. This might, in return, provide some insights for the hand-crafted architecture design.

## Acknowledgement

This work is supported by Major Project for New Generation of AI Grant (No. 2018AAA0100402), National Key R&D Program of China (No. 2017YFA0700800), and the National Natural Science Foundation of China under Grants 61976208, 91646207, 61573352, and 61773377. This work is also supported by Beijing Academy of Artificial Intelligence (BAAI).

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
