[Supplementary Material]

## Supplementary Material

### Algorithms

DetNAS consists of three steps: supernet pre-training on ImageNet, supernet fine-tuning on detection datasets and architecture search on the trained supernet with EA. We formulate them in algorithms here. The first two steps are combined into Algorithm 1. The third step is formulated in to Algorithm 2.

| **Algorithm 1:** SuperNet Training | **Algorithm 2:** Search on Supernet with EA |
|---|---|
| **Input** : Search space $\mathbb{S}$, Detector $\mathcal{D}$, ImageNet pre-training iterations $I_P$, Detection fine-tuning iterations $I_F$, ImageNet pre-training dataset $\mathbb{T}_{Pre}$, Detection fine-tuning dataset $\mathbb{T}_{Det}$. | **Input** : Trained supernet model $S$, Detector $\mathcal{D}$, Population size $|\mathbf{P}|$, Parent size $|\mathcal{P}|$, Detection validation set $\mathbb{V}_{Det}$, Small subset of detection training set $\mathbb{T}_{Det_S}$, Evolution iterations $I_E$, Constraint $\eta$. |
| 1 $S \leftarrow$ `initialize`$(\mathbb{S})$ <br> 2 $i \leftarrow 0$ <br> 3 **while** $i < I_P$ **do** <br> 4 $\quad \theta \leftarrow$ `random-path`$(\mathbb{S})$ <br> 5 $\quad S \leftarrow$ `training`$(S(\theta),\ \mathbb{T}_{Pre})$ <br> 6 $\quad i \leftarrow i + 1$ <br> 7 $j \leftarrow 0$ <br> 8 **while** $j < I_F$ **do** <br> 9 $\quad \theta \leftarrow$ `random-path`$(\mathbb{S})$ <br> 10 $\quad S \leftarrow$ `training`$(\mathcal{D}(S(\theta)),\ \mathbb{T}_{Det})$ <br> 11 $\quad j \leftarrow j + 1$ | 1 $\mathbf{P}^{(0)} \leftarrow$ `random-initialize`$(|\mathbf{P}|, \eta)$ <br> 2 $f_{best}, \theta_{best} \leftarrow (0, \varnothing)$ <br> 3 **for** $k \in (1\ to\ I_E)$ **do** <br> 4 $\quad$ **for** $P \in \mathbf{P}^{(k)}$ **do** <br> 5 $\quad\quad S(P.\theta) \leftarrow$ `getBN`$(\mathcal{D}(S(P.\theta)),\ \mathbb{T}_{Det_S})$ <br> 6 $\quad\quad P.f \leftarrow$ `evaluate`$(\mathcal{D}(S(P.\theta)),\ \mathbb{V}_{Det})$ <br> 7 $\quad\quad$ **if** $P.f > f_{best}$ **then** <br> 8 $\quad\quad\quad f_{best}, \theta_{best} \leftarrow (P.f, P.\theta)$ <br> 9 $\quad \mathcal{P} \leftarrow$ `select-topk`$(\mathbf{P}^{(k)},\ |\mathcal{P}|)$ <br> 10 $\quad \mathbf{P}^{(k+1)} \leftarrow$ `mutate-crossover`$(\mathcal{P},\ \eta)$ |
| **Output** : The trained supernet model $S$ | **Output** : The best architecture $\theta_{best}$ |

### Comparisons in the "2x" Settings

Table 1: Main result comparisons in "2x" settings

| Backbone | ImageNet Classification | | Object Detection with FPN on COCO | | | | | |
|---|---|---|---|---|---|---|---|---|
| | FLOPs | Accuracy | mAP | $AP_{50}$ | $AP_{75}$ | $AP_s$ | $AP_m$ | $AP_l$ |
| ResNet-50 | 3.8G | 76.15 | 39.3 | 60.3 | 42.9 | 23.9 | 41.8 | 51.9 |
| ResNet-101 | 7.6G | 77.37 | 40.9 | 61.9 | 44.9 | 24.2 | 43.8 | 54.0 |
| ShuffleNetv2-40 | 1.3G | 77.18 | 41.1 | 62.6 | 45.4 | 24.6 | 44.2 | 54.2 |
| ShuffleNetv2-40 (3.8) | 3.8G | 78.47 | 42.4 | 63.6 | 46.7 | 26.2 | 45.5 | 55.6 |
| DetNASNet | 1.3G | 77.20 | 41.8 | 63.3 | 45.5 | 25.4 | 44.8 | 55.1 |
| DetNASNet (3.8) | 3.8G | 78.44 | **43.4** | 64.9 | 47.3 | 25.9 | 46.7 | 58.0 |

\* Results are trained with the same setting as in Section 4, except that the training setting is "2x" in Detectron.

Results in the paper are trained with the "1x" setting in Detectron to keep consistency with the supernet training. Here we report the comparisons in "2x" setting. DetNASNet and DetNASNet (3.8) are still superior to the hand-crafted ResNet-50, ResNet-101 and ShuffleNetv2-40.

### Inference time comparisons

Table 2: Inference and mAP of ResNet and DetNASNet on FPN.

| | ResNet-50 | ResNet-101 | ShuffleNetv2-40 | ShuffleNetv2-40 (3.8) | DetNASNet | DetNASNet (3.8) |
|---|---|---|---|---|---|---|
| FPS | 17.9 | 15.3 | **21.8** | 17.2 | 20.4 | 15.8 |
| $mAP_{1\times}$ | 37.3 | 40.0 | 39.2 | 40.8 | 40.2 | **42.0** |
| $mAP_{2\times}$ | 39.3 | 40.9 | 41.1 | 42.4 | 41.8 | **43.4** |

\* We measure the inference time on Tesla V100 and our platform Brain++ with input size (800, 1200).

Although inference time is not the target of this work, we measure the FPS to avoid the concern about the speed of DetNASNet. DetNASNet processes 5 more frames per second than ResNet-101. DetNASNet (3.8) is only 2 FPS slower than ResNet-50 but has a much better mAP.