[Reviews · NeurIPS 2019]

Reviewer 1



This paper proposes a neural network search strategy for object detection task. The problem is interesting and useful for many real applications. This paper gives a three stage solution that can search pre-training based detectors effectively and efficiently. Experiments on both COCO and VOC are conducted to show the effectiveness of the proposed solution, and detection based models are superior than classification based models. The idea of searching network structure for detection with pre-training stage is novel and interesting. Firstly, the authors give detailed introduction about existing NAS related works. Then NAS for object detection naturally becomes an important task to solve. And state-of-the-art detection models are usually pre-trained on large classification datasets such as ImageNet. So the authors give a three steps pipeline to solve this problem. The paper is written clearly and easy to understand. And the solution is clever and efficient. It only takes 44 GPU days (twice of training a normal detector) to finish the DetNAS on COCO. There are also some concerns about this paper. (1) In order to reduce the training instability, evolutionary based solution is adopted to generate new models. Since model search space is really large, how to sample in this space is crucial for the final performance. Will the evolutionary solution be sub-optimal? From the experiments, we can see that the best one of 5 random models can achieve pretty close results (less than 1 mAP) compared to the carefully searched model. And it’s interesting to see the performance curve of using different number of random models such as 1,5,10,20 and so on. (2) Although the search space is claimed to be “large” already, actually only four candidates are supported for each block. Other choices such as Pooling positions, channel size, complex connections are not considered in this paper. (3) Shufflenet block based models are efficient on CPU and ARM platforms, however, the latency of shufflenet is supposed to be much higher than resnet under same model complexity (FLOPS). Post Rebuttal: The rebuttal has answered my questions. I will keep my score.

Reviewer 2



Strengths: 1. This paper first proposes a framework for object detection backbone search. 2. The experiments show improvement between DetNAS and hand-crafted networks on COCO and VOC detection datasets. 3. Code is available. Weaknesses: 1. The comparison between DetNAS and ResNet is unfair since ShuffleNetv2 block is more efficient than the Residual block in ResNet. I suggest the authors provide more comparison between DetNas and other networks constructed by effitive block. For example, the author should provide the result of a 3.8G FLOPs ShuffleNetv2 in Table 2. 2. The novelty of this paper is limited. The one-shot search mothod is first proposed by [1]. An interesting result is that the improvement between DetNAS and Random structure is minor. It seems a carefully designed search space is more important than the search method. The authors shold provide the mmAP curve of parents network during EA search. Ref: [1] Zichao Guo, Xiangyu Zhang, Haoyuan Mu, Wen Heng, Zechun Liu, Yichen Wei, and Jian Sun. Single path one-shot neural architecture search with uniform sampling. abs/1904.00420, 2019.

Reviewer 3



Summary: This work uses existing supernet based one-shot architecture search methods and uses them to search for backbones for the object-detection. The paper motivates the need for searching backbone architectures by noticing that backbone architectures used for object detection tasks are mostly off-the-shelf classification networks and may not be the best for object detection. The authors use ShuffleNet v2[1] based block and use a one-shot supernet search technique similar to the one proposed in [2]. The paper used an Evolutionary algorithm to search for child networks in the supernet. The authors conduct experiments using a one-stage detector (RetinaNet) and a two-stage detector (FPN) and make the following observations: a) The searched architecture performs better than the ResNet and ShuffleNet based backbones while consuming fewer FLOPs. b) The architectures optimized for detection perform better than the ones optimized for classification. c) There are qualitative differences between architectures optimized for classification as opposed to detection. Originality: The paper has used a previously known supernet based one-shot architecture search technique and applies it to the domain of object detection. For the purpose of object detection, the technique makes the necessary change in the formulation to initialize the weights from a classification task. The paper has noticed qualitative differences between architectures optimized for detection v/s classification as found by their search method. Quality: The paper has conducted the experiments necessary to show the benefit of their technique. The ablation studies also help identify the performance impact due to optimizing for classification tasks and the effect of the search space. Compared to time taken train modern object detectors, the search method is also very computationally efficient taking only 20 GPU-days. Although the effect of searching an architecture is clear, there exist much better mAP numbers in literature with hand-tuned architectures. For example, the best 2 stage mAP in the paper is 42.0 whereas [3] achieve 45.1 with an hourglass backbone and test-time augmentations. This limits the utility of this approach. The paper also does not report FPS achieved by their final architecture, which is an important consideration for many applications. Another limitation is that the backbone search-space by itself was designed for the classification task which may limit the final performance. Clarity: The paper is well-written for the most part. Sections 3.2 and 4 can use some work to make what is being done more clear. Particularly, the details of how the final search on the trained supernet is performed is not clear by reading the paper itself. The detail that the child networks have to be only trained by updating batch-norm statistics is not emphasized enough and requires careful reading of the algorithm in supplementary. The experiments and ablation studies themselves are explained well and it is clear what is being measured. Significance: This work shows that the common practice of pre-training detection backbones for classification is sub-optimal and deserves to be revisited. The paper shows that one-shot supernet based methods are promising research directions to find better detection backbones. If the detection algorithms used were better, the proposed technique could be more useful for practitioners in the field as they could just use the found architecture off the shelf, which they currently would not prefer doing since the mAP performance on datasets are sub-optimal. References: [1]: N. Ma, X. Zhang, H.-T. Zheng, and J. Sun. Shufflenet v2: Practical guidelines for efficient cnn architecture design. In Proceedings of the European Conference on Computer Vision (ECCV), pages 116–131, 2018. 5, 7 [2]: Zichao Guo, Xiangyu Zhang, Haoyuan Mu, Wen Heng, Zechun Liu, Yichen Wei, and Jian Sun. 315 Single path one-shot neural architecture search with uniform sampling. abs/1904.00420, 2019. [3]: X.Zhou, D.Wang, and P.Krahenb ¨ uhl. Objects as points. ¨arXiv preprint arXiv:1904.07850, 2019. 1 Post-Rebuttal: The rebuttal has conducted experiments to address my major concerns. It is clear that the algorithm is competitive with modern detection approaches while not compromising on speed.

Reviewer 4



Considering this is the first work to adopt NAS on searching the backbone CNN for the detection task and the proposed solution is clearly better than randomly generated networks. I would suggest a weakly accept for this work. I do not suggest score better than a weakly accept due to the following concerns: Main concern: 1) The effectiveness of DetNAS is mainly benefited from better search space rather than the proposed searching method. The proposed solution does not significantly better than randomly generated networks. As can be seen in Table 4 and Table 3, the “Random” Network is already significantly better than the baseline ShuffleNetV2. However, the gap between “Random” Network and the proposed DetNAS is marginal. Minor concern: 2) Considering the search space is inspired by ShuffleNetV2 rather than ResNet. Only emphasizing the superiority of proposed DetNAS over ResNet in abstract is inappropriate and misleading. BTW, next time, please remember to remove the .git folder from your code to follow the double-blind peer review rule. Otherwise, the author information might be leaked.

[Author Response · NeurIPS 2019]

We sincerely thank four reviewers for the constructive comments. We have conducted experiments to address almost all questions. As the issue about random models is concerned by Reviewer #1, #2 and #4, we answer this first.

Figure R-1. Random models on FPN in small and large search space.      Figure R-2. mmAP during search for main results.

**Q:** *Issues on the random models.* **A:** To clarify this issue, three additional experiments have been conducted:
**(1)** As in Figure R-1 (left), 15 additional random models are sampled and trained in the small search space (20 in total, with the original 5 models in the paper). We depicted the scatter and the average line of random models and the line of DetNAS. DetNAS in the small search space is 36.4% while the Random is 35.6±0.6%. **(2)** As in Figure R-1 (right), 7 random models are sampled and trained in the large search space (The maximum number of models we can train in this week). DetNAS in the large search space is 40.0 while Random is 38.8% (+1.2%) in average with the highest 39.1% (+0.9%), which is a large margin. **(3)** As in Figure R-2, the mmAP curve on the supernet search are depicted to compare EA with Random. For each iteration, top 50 models until the current iteration are depicted at each iteration. EA demonstrates a clearly better sampling efficiency than Random.

These comparisons show that: **(1)** In the large search space for main results, it is difficult to randomly pick a suitable model. The improvement is not marginal (+1.2%/0.9%). **(2)** In the small search space, DetNAS is clearly better than Random while the improvement is not large enough as constrained by the small search space. **(3)** As in Figure R-2, EA is much more sampling efficient than Random search. Thanks the reviewers for the constructive suggestions on this issue. We hope that these experiments can clarify the concerns. These experiments will be included in the next version.

**For Reviewer #1**

**Q:** *Inference time.* **A:** For the concern on the inference, we measure the FPS of DetNASNet and ResNet on the same Tesla V100 with (800,1200) input size, as in Table R-1. Under the same mAP, DetNASNet processes 5 more frames per second than ResNet-101. Under the same FLOPs, DetNASNet (3.8) is only 2 FPS slower than ResNet-50 but has a much better mAP (42.0 vs 37.3). The latency of DetNASNet is not much higher than ResNet under the same FLOPs.
**Q:** *More random models.* **A:** Thanks for the beneficial suggestion. More random models are provided as above. The improvement is constrained by the small search space. In the large search space, the gap is clear (+1.2%/0.9%).

Table R-1. Inference and mAP of ResNet and DetNASNet on FPN.

|  | ResNet-50 | ResNet-101 | DetNASNet | DetNASNet (3.8) |
|---|---|---|---|---|
| FPS | 17.9 | 15.3 | **20.4** | 15.8 |
| $\text{mAP}_{1\times}$ | 37.3 | 40.0 | 40.0 | **42.0** |

Table R-2. Comparison with the 3.8G ShuffleNetv2.

|  | ImageNet | $\text{mAP}_{1\times}$ | $\text{mAP}_{2\times}$ |
|---|---|---|---|
| ShuffleNetv2 | **78.47** | 40.8 | 42.4 |
| DetNASNet (3.8) | 78.44 | **42.0** | **43.4** |

**For Reviewer #2**

**Q:** *Comparison with a 3.8G FLOPs ShuffleNetv2.* **A:** Thanks for this suggestion. In Table R-2, the results of a 3.8G FLOPs ShuffleNetv2 are provided and compared with 3.8G DetNASNet in both 1x and 2x training settings on FPN.
**Q:** *mmAP curve of during EA search.* **A:** The search curve of EA against random search is shown in Figure R-1. It shows top 50 models in each iterations. Parent models are top 10 among them. EA is more efficient than Random.

**For Reviewer #3**

**Q:** *A more general search space that includes Hourglass.*
**A:** Designing a search space that includes Hourglass is a promising and interesting idea. We would like to try it in the future work.

Table R-3. Comparison with CenterNet+Hourglass.

|  | n.a | +flip | +multi scale |
|---|---|---|---|
| CenterNet-Hourglass104 | 40.3 | 42.2 | 45.1 |
| FPN-DetNASNet (3.8) | 43.4 | 44.8 | **46.1** |

**Q:** *The practical utility of their methods.* **A:** In terms of accuracy, we obtain 46.1 mmAP by simply following the test augmentations used by CenterNet as in Table R-3. CenterNet is one-stage while Hourglass104 has much more FLOPs than 3.8 G. In terms of FPS, DetNASNet has a clear superiority as in Table R-1. Better detection strategies are important for the practical utility. The focus of this paper is backbone search, thus we leave this in the future work.
**Q:** *Writing.* **A:** Thanks for your suggestions on writing. We will revise the paper thoroughly under your suggestions.

**For Reviewer #4**

**Q:** *The gap between Random and DetNAS.* **A:** Thanks for pointing out the concern of random models. We provide the search curve and more results of random models in both the small and large search space. The reason for the small improvement mainly comes from that the search space for the ablation studies is not large enough and the final results trained are close. In the large search space, the improvement is clear (+1.2%/0.9%) as shown in Figure R-1 (right).
**Q:** *Minors.* **A:** Thanks for your suggestions. We will revise the abstract for this point carefully in the next version.

[Meta-Review · NeurIPS 2019]

Thanks for the paper submission and for addressing important concerns in the author feedback. We believe this is a valuable contribution to NeurIPS. The main contribution on backbone search for object detection is of interest. This paper is clear, with interesting insights and applicability. We appreciate the extensive experimental evaluation. As areas of improvement, we strongly recommend the incorporation of the insights in the author feedback into the camera-ready version of the paper.